# Ferroptosis in Non-Small Cell Lung Cancer: Progression and Therapeutic Potential on It

**DOI:** 10.3390/ijms222413335

**Published:** 2021-12-11

**Authors:** Jiayu Zou, Li Wang, Hailin Tang, Xiuxiu Liu, Fu Peng, Cheng Peng

**Affiliations:** 1State Key Laboratory of Southwestern Chinese Medicine Resources, School of Pharmacy, Chengdu University of Traditional Chinese Medicine, Chengdu 611137, China; vincent@stu.cdutcm.edu.cn; 2Department of Pharmacology, Key Laboratory of Drug-Targeting and Drug Delivery System of the Education Ministry, Sichuan Engineering Laboratory for Plant-Sourced Drug and Sichuan Research Center for Drug Precision Industrial Technology, West China School of Pharmacy, Sichuan University, Chengdu 610041, China; wangli123ai@163.com (L.W.); liuxiuxiu@scu.edu.cn (X.L.); 3Department of Breast Oncology, Sun Yat-sen University Cancer Center, Guangzhou 510275, China; tanghl@sysucc.org.cn

**Keywords:** ferroptosis, NSCLC, progression, therapy resistance, natural drug

## Abstract

As a main subtype of lung cancer, the current situation of non-small cell lung cancer (NSCLC) remains severe worldwide with a 19% survival rate at 5 years. As the conventional therapy approaches, such as chemotherapy, radiotherapy, targeted therapy, and immunotherapy, gradually develop into therapy resistance, searching for a novel therapeutic strategy for NSCLC is urgent. Ferroptosis, an iron-dependent programmed necrosis, has now been widely considered as a key factor affecting the tumorigenesis and progression in various cancers. Focusing on its effect in NSCLC, in different situations, ferroptosis can be triggered or restrained. When ferroptosis was induced in NSCLC, it was available to inhibit the tumor progression both in vitro and in vivo. The dominating mechanism was due to a regulation of the classic ferroptosis-repressed GSH-dependent GPX4 signaling pathway instead of other fractional regulating signal axes that regulated ferroptosis via impacting on the ROS, cellular iron levels, etc. In terms of the prevention of ferroptosis in NSCLC, an GSH-independent mechanism was also discovered, interestingly exhibiting the same upstream as the GPX4 signaling. In addition, this review summarizes the progression of ferroptosis in NSCLC and elaborates their association and specific mechanisms through bioinformatics analysis with multiple experimental evidence from different cascades. Finally, this review also points out the possibility of ferroptosis working as a novel strategy for therapy resistance in NSCLC, emphasizing its therapeutic potential.

## 1. Introduction

Hundreds of years ago, researchers discovered that cell death happened naturally. With the development of science and technology, programmed cell death (PCD) was finally developed, explaining this kind of “cell suicide”, and it had participated in the progression of abundant diseases and development of cancers [1]. PCD mainly includes apoptosis, autophagy, and programmed necrosis, and it is a definition in the opposite of an accidental type of cell death that is named cell necrosis. In addition, the programmed necrosis contains different cell death types such as pyroptosis, necroptosis, and ferroptosis, etc. [2,3]. Apart from apoptosis, the programmed necrosis occurs with inflammation. Pyroptosis is a type of cell death depending on human caspase 1, 4, 5, and 12, etc. However, the necroptosis and ferroptosis are caspase-independent [4,5]. As a type of programmed necrosis, ferroptosis is mediated by phospholipid peroxidation together with free iron-mediated Fenton reactions, which means that ferroptosis is an iron-dependent cell death different from apoptosis [6,7] (Figure 1).

To provide solid evidence on the reliability of the inner connection and significance for ferroptosis in non-small cell lung cancer (NSCLC), we summarize the bioinformatic studies, specific mechanisms, and therapeutic values of ferroptosis in NSCLC in the past five years based on PubMed, web of science, Springer, and Wiley database searches. Duplicate articles and articles with similar results were excluded. Only 99 articles were finally included after reading the titles, abstracts, and whole papers (Figure 2).

## 2. Ferroptosis in Cancer

### 2.1. Ferroptosis Was Observed in Diverse Cancers

Over the past few years, ferroptosis was observed in many types of cancers, participating in its progression, being connected with the therapeutic response, and working as a potential strategy against different therapy resistance [8] (Figure 3) For example, Schmitt et al. elucidated that the therapeutic efficacy of dimethyl fumarate in diffuse large B-cell lymphoma was mainly due to an induction of ferroptosis [9]. Otherwise, in gastric cancer, Zhang et al. demonstrated the repressed status of ferroptosis was a promoter to the acquired chemotherapy-resistance [10]. Meanwhile, the sorafenib resistance in hepatocellular carcinoma could also be augmented by restrained ferroptosis [11].

### 2.2. Ferroptosis Could Be Triggered by Autophagy in Cancer

As a special type of cell death, autophagy had been proven in connection with the PCD and cell survival [12], and Emdad et al. revealed the existence of a common upstream link between autophagy and apoptosis in human high-grade gliomas [13]. Hence, it hinted at the association between autophagy and ferroptosis in cancers. 

Most recently, Kremer et al. observed ferroptosis-related changes in cysteine, glutathione (GSH), and lipid antioxidant when blocking the cytosolic aspartate aminotransaminase (GOT1) in pancreatic cancer. The underlying mechanism was deciphered to be a repressed status of mitochondrial metabolism and an increased catabolic state that resulted from the GOT1-silencing, and accounting for autophagy. Later, this abnormal status activated the labile iron availability, thus activating the ferroptotic stimuli and triggering ferroptosis [14]. Otherwise, Gao et al. pointed out “ferritinophagy” and demonstrated the influences caused by autophagy on the cellular iron storage protein that came to an autophagic degradation in ferroptosis [15]. Correspondingly, vitamin C was also illustrated to be an induction on ferroptosis through this ferritinophagy activation in anaplastic thyroid cancer cells [16]. 

In addition, other research also explained that, in gastric cancer [17], breast cancer [18], and NSCLC [19], autophagy could act as a ferroptosis inducer through diverse mechanisms. 

## 3. Mechanism of Ferroptosis in NSCLC

### 3.1. Brief Background of NSCLC

According to the most recent data from the Global Cancer Observatory (GCO) database, lung cancer ranked as the third highest incidence rate (22.4 per 100,000 population) and the highest fatality rate (18 per 100,000 population) worldwide in 2020. NSCLC is the main subtype of lung cancer, with an approximate ratio of 85% [20]. Additionally, the prognosis of NSCLC remains poor, with a 19% survival rate at 5 years, necessitating the development of an efficient treatment approach [21]. During the past decades, diverse therapeutic strategies such as surgery, chemotherapy, radiotherapy, targeted therapy, and immunotherapy were applied into the clinical NSCLC treatment [22]. Among them, surgery was suitable for the NSCLC patients in an early stage [23]. While chemotherapy was mostly dependent on antitumor platinum drugs as the clinical first-line treatment in NSCLC [24], radiotherapy relied on the application of radioactive rays. As for immunotherapy, standard regimens mainly focused on the immune checkpoint inhibitors, such as Keytruda and Opdivo, which worked against PD-1 or PDL-1, thereby activating the body immune system [25,26,27]. Additionally, the targeted therapy for NSCLC specifically aimed *EGFR*, *ALK*, *KRAS*, etc. to change the relative biology process, showing a great value [28].

Despite the extraordinary effects of these therapy strategies, NSCLC inevitably developed into therapy resistance to some degree [29,30]. Inspiringly, an abundance of neoadjuvant therapies and combined therapies were developed, and could at least partly overcome this kind of resistance. Usually, it was a combination of the therapies mentioned above, becoming diverse clinical regimens [31,32]. As for the targeted therapy, plenty of next generation tyrosine kinase inhibitors (TKIs) were found and used in clinics, avoiding therapy resistance to some extent [33,34]. Moreover, a lot of natural or chemical drugs were also observed to have a synergistic effect with chemotherapy or radiotherapy when they were applied together, offering another potential therapeutic strategy [35,36]. 

Ferroptosis has now been found in various types of lung injuries or diseases, such as PM2.5-induced lung injury, lung epithelial injury, and acute lung injury [37,38,39,40]. In particular, ferroptosis was observed to have a crucial role in affecting lung cancer [41]. However, based on the huge proportion of NSCLC in lung cancer, recent studies have centered on its effects on NSCLC. Hence, it is worthwhile to excavate the inner relevance and mechanisms of ferroptosis in NSCLC.

### 3.2. Current Discoveries on the Mechanisms of Ferroptosis in NSCLC

Although ferroptosis was widely studied in various diseases or cancers, the progression of it in NSCLC have not been specifically summarized to date. Among all the up-to-date studies concerning to ferroptosis in NSCLC, several key factors, such as glutathione peroxidase 4 (*GPX4*), reactive oxygen species (ROS), solute carrier family 7, (cationic amino acid transporter, y+ system) member 11 (*SLC7A11*), subcellular localization of nuclear factor E2-related factor 2 (*NRF2*), and cystine/glutamate transporter (*xCT*), etc., were observed to have a part in this process, deciphering various mechanisms (Table 1).

The current mechanism of ferroptosis in NSCLC including GPX4 pathway has been concluded, revealing the mechanism of the depleted cysteine induced ferroptosis in NSCLC. Most interestingly, a new discovery on the GSH-independent pathway was found in NSCLC during the summarization process and the decreased cysteine suppressed ferroptosis without affecting GSH. Additionally, other relative studies were also introduced as below. 

The GSH-dependent GPX4 reductive system is one of the most essential ferroptosis-repressed pathways [63]. This signaling pathway can be initiated by cystine when the System Xc^-^ is stimulated by *P53* or erastin [64]. The System Xc^-^ contains protein *SLC7A11* and (solute carrier family 3 (activators of dibasic and neutral amino acid transport), member 2) *SLC3A2*; it is able to achieve the intracellular exchange of cystine and glutamate. When the cystine enters the cell, it will be oxidized into cysteine. Then, the cysteine combines with the cellular glutamate into γ-Glutamylcysteine (GGC) through a modification of the glutamate-cysteine ligase (GCL). Later, with the help of glutamate synthetase (*GSS*) and Glycine (Gly), the GGC is changed into GSH, which plays a crucial role in the cellular peroxidation and is implicated in the GPX4 bioprocess, decreasing its expression level. In short, when the cystine or cysteine is silenced, the cellular ROS will increase, resulting in ferroptosis [8,65] (Figure 4A). 

Consistent with this, evidence elucidated that the induction of ferroptosis in NSCLC was associated with this mechanism through an increased ROS level. Alvarez et al. found that suppressing the (Cysteine desulfurase NFS1) *NFS1*, an iron-sulfur cluster biosynthetic enzyme, would affect the primary lung tumor growth. Mechanistically, *NFS1*-silencing led to an insufficiency of iron-sulfur cluster maintenance, which later inhibited the GSH biosynthesis through iron-starvation response, and along with the suppression of cysteine transport, finally triggering ferroptosis in lung adenocarcinomas. Additionally, it restrained tumor growth in vivo [66]. 

Concentrating on the cysteine, however, the downregulated level of cysteine could also prevent ferroptosis without impairing the GSH in NSCLC. Kang et al. found that γ-cysteine ligase catalytic subunit (*GCLC*) had the activity to upregulate γ-glutamyl-peptides levels when cysteine was silenced or there was an overexpression of *NRF2* in NSCLC cells. Later, the γ-glutamyl-peptides played a crucial role in decreasing the glutamate accumulation to inhibit ferroptosis. Notably, when cysteine was blocked, *GCLC* suppressed ferroptosis rather than triggering it. Most importantly, this process occurred without the participation of GSH by protecting the glutamate homeostasis. This meant that *GCLC* worked as a non-canonical glutamate-cysteine ligase in suppressing ferroptosis in NSCLC and indicated a different pathway from the classic GSH-dependent GPX4 pathway in NSCLC when the cysteine was silenced [67] (Figure 4B).

Intriguingly, focusing on the *NFR2*, a one-year early study revealed its mechanism in NSCLC. NFR2 participated in the regulatory process of ferroptosis when cysteine was downregulated. Liu et al. elucidated that NRF2 could negatively regulate (Focadhesin) *FOCAD*, depending on the RPA1-ARE complex. However, FOCAD could only make the NSCLC cells sensitive to ferroptosis that was induced by cysteine-silencing rather than *GPX4*-inhibition. In addition, it was the (focal adhesion kinase) *FAK* that was activated by FOCAD, being able to activate the TCA cycle and the complex I (NADH) activation in mitochondrial electron transport chain (ETC). Thus, it sensitized NSCLC cells to ferroptosis that was induced by cysteine-silencing eventually. Moreover, experiments in vivo and in vitro both confirmed a better therapeutic effect when brusatol, the NRF2 inhibitor, was treated with erastin rather than being used singly. Above all, it verified that the cysteine-induced ferroptosis was linked to NFR2 and FOCAD-FAK signal in NSCLC patients [68]. Taken together, there was a clue on the possible connection between FOCAD-FAK signal and the GCLC activation. Moreover, as they both could lead to an inhibition of ferroptosis in NSCLC, whether the FOCAD-FAK signal worked synergistically with the glutathione-independent pathway deserves further exploration. 

In addition, the downregulation of cystine acted the same effect with cysteine. Poursaitidis et al. confirmed that the cystine-silencing in human mammary epithelial (HME) cells promoted the occurrence of EGFR-mutation and ferroptosis with an elevated ROS level. Otherwise, when EGFR was silenced, a further inhibition on ferroptosis happened correspondingly, indicating a major role of the EGFR-mutation in the cystine-silencing-induced ferroptosis. Then, treatment with a cystine-depleting enzyme in the EGFR mutant NSCLC xenograft mice confirmed the results in vivo. An inhibition on the tumor growth was observed, implying the correlation in ferroptosis and EGFR-mutant NSCLC [69].

## 4. Specific Evidence Elucidating the Connection between Ferroptosis and NSCLC

### 4.1. Bioinformatics Implying the Relationship among Ferroptosis, NSCLC, and Clinical Immunotherapy

Bioinformatics is a novel analysis technology widely used for the prediction of outcomes in cancer immunotherapy [70]. Evidence also highlighted a possible association between ferroptosis and immunity-related genes in NSCLC with the advancement of bioinformatics in pharmaceutical studies. In recent times, Zhang et al. discovered the link between ferroptosis-related genes and LUAD, a subtype of NSCLC, revealing the predictive significance of 15 ferroptosis genes (*RELA*, *ACSL3*, *YWHAE*, *EIF2S1*, *CISD1*, *DDIT4*, *RRM2*, *PANX1*, *TLR4*, *ARNTL*, *LPIN1*, *HERPUD1*, *NCOA4*, *PEBP1*, and *GLS2*) in clinical NSCLC patients. Researchers took gene datasets from The Cancer Genome Atlas (TCGA), GEO, and FerrFb database, and then filtered them through the Kaplan–Meier and univariate Cox analysis for prognostic potential. Later, the least absolute shrinkage and selection operator (Lasso) Cox regression model and receiver operating characteristic (ROC) analysis were also used. Finally, gene set enrichment analysis (GSEA) and immune infiltration analyses identified the function of 15 ferroptosis-related genes with specific pathways in resting mast cells and resting dendritic cells, exerting the prognosis value of the 15-gene signature [71].

Using a univariate and multivariate Cox regression analysis for RNA-sequencing data and clinical information from the TCGA database, Liu et al. also found the ferroptosis-related gene in NSCLC. After that, a ROC model was used to assess the sensitivity and specificity, and differentially expressed genes were discovered using gene ontology enrichment and Kyoto Encyclopedia of Genes and Genomes (KEGG) pathway analyses. Consequently, five ferroptosis-related genes (*FANCD2*, *GCLC*, *SLC7A11*, *ALOX15*, and *DPP4*) were discovered to be expressed differentially and to have an impact on immune-related pathways in NSCLC, confirming a putative correlation between ferroptosis and NSCLC and revealing new insights on ferroptosis for NSCLC clinical immunotherapy [72].

Han et al. also collected NSCLC-relative gene expression data from public databases. Later, the common intersections between ferroptosis-related genes and differential expressed genes (DEGs) were chosen as the candidate ferroptosis-related genes, and DEGs were from the cancer tissues and Para cancerous tissues. Furthermore, bioinformatic analysis approaches such as univariate Cox analysis and Lasso analysis were used to split the samples into high-risk and low-risk groups, resulting in a prognostic risk-score model. Additionally, Single-sample GSEA (ssGSEA), Kaplan–Meier curve, ROC curve, Univariate, and multivariate Cox analyses were also used to confirm early prediction of this model. The results revealed that the candidate ferroptosis-related genes in the high-risk and low-risk groups were not only enriched in pathways that appeared to be related to immunosuppressive status, but they also had the potential to become prognostic factors that improved NSCLC patients’ overall survival time [73].

### 4.2. Non-Coding RNA Works as a Bridge Connecting Ferroptosis and NSCLC

Non-coding RNA was a type of transcriptomes that usually did not participate in the coding process of proteins [74]. Thereby, it was considered to have no effect on the biological process for a long time, even recognized as a meaningless noise or “dark matters” [75]. However, throughout the last two decades, several studies have been conducted to understand the various bioactivities of non-coding RNA, with a focus on the three primary subtypes: microRNA, long non-coding RNA (LncRNA), and circular RNA (circRNA). Accumulating evidence indicated that these non-coding RNA played an important role in the initiation and development of NSCLC [76,77,78,79,80,81]. Recent studies have also given insight into the particular impact of distinct microRNAs and lncRNAs in NSCLC via ferroptosis modulation.

#### 4.2.1. Micro-RNA

The expression levels of lipid peroxidation and intracellular iron were measured in NSCLC cells that were treated with erastin or RSL3. Then, the group incubated with miR-302a-3p mimic was found an elevation of lipid peroxidation and cellular iron, which implied an inductive effect of miR-302a-3p on ferroptosis. Correspondingly, the miR-302a-3p inhibitor group also showed a suppressive effect on the erastin- or RSL3-related-ferroptosis. Furthermore, miR-302a-3p could directly bind to the 3′-UTR regions of ferroportin. Therefore, the miR-302a-3p was able to downregulate the expression of ferroportin, which consequently triggered ferroptosis in NSCLC [42].

Interestingly, on the flip side, microRNA could also suppress ferroptosis in NSCLC. Song et al. implemented the bioinformatics analysis and luciferase assay uncovering a directly binding of miR-4443 with its downstream gene METTL3. Further, mechanistic analysis revealed that the upregulation of miR-4443 could regulate FSP1, a protein that suppressed ferroptosis, in an m6A manner when NSCLC cells were exposed to cisplatin in vitro. Moreover, due to the combination of miR-4443 and METTL3, it aggravated the occurrence of chemotherapy resistance. In addition, the consequence was also consistent when miR-4443 was upregulated in vivo, leading to an increase in the growth of tumors [43]. 

#### 4.2.2. LncRNA

As for lncRNA, both lncRNA NEAT1 and MT1DP worked a promoter action on the erastin-induced ferroptosis in NSCLC cells [45,46]. 

LncRNA NEAT1 was reported as having an impact on the expression of *ASCL4*, *SCL7A11*, and *GPX4*. Wu et al. performed relative experiments verifying the relevance of ferroptosis and lncRNA NEAT1 in NSCLC in vitro. The combination of LncRNA NEAT1 and *ACSL4* was elucidated by the dual-luciferase assay. Then, TUNEL staining and malondialdehyde levels were detected. The results of these assays, including rescue experiments, confirmed the regulatory effect of LncRNA NEAT1 on ferroptosis. Finally, the specific protein expression levels in different erastin concentration groups were also assessed by Western blot assay. Overall, when lncRNA NEAT1 was silenced in erastin-treated NSCLC cells, it would affect ferroptosis more reliant on the *ASCL4* [45].

LncRNA MT1DP had a suppression on antioxidation, research found that there was a connection of MT1DP and *NRF2*. The MT1DP overexpression in erastin-treated NSCLC cells could affect miR-365a-3p, and then it came with a downregulation of antioxidant transcript factor *NRF2*. In addition, ferroptosis-related molecules such as Fe^2+^, ROS, and malondialdehyde (MDA) levels were also elevated. However, the GSH level was decreased at the same time. Furthermore, when using FA-LP nanoparticles as a co-delivery system to wrap erastin and MT1DP for a targeted therapy, tests revealed that it not only downregulated MT1DP function in the opposite direction, but also had a synergistic impact. Moreover, FA-LP nanoparticles made NSCLC cells more vulnerable to erastin-induced ferroptosis in vitro, as evidenced by decreased GSH levels and increased lipid ROS. Additionally, xenograft models were also established to verify its availability [46].

Wang et al. proved that there was a high expression level of lncRNA LINC00336 in lung cancer, and it could bind *ELAVL1*, resulting in an inhibition on ferroptosis. In turn, *ELAVL1* had an effect on the posttranscriptional level of LINC00336, leading to overexpression [47]. Otherwise, as the MIR6852 was positively correlated with ferroptosis, LINC00336 and MIR6852 had a competing endogenous RNA interaction. Endogenous RNA competing with microRNA response elements (MREs) was a notion, stating that MREs served as a link between different types of RNA transcripts [82]. As LINC00336 could sponge MIR6852, affecting the cellular cystathionine-beta-synthase (CBS) expression level. In a different way, it validated the inhibitory activity of LINC00336 on ferroptosis [47].

### 4.3. Various Drugs Show A Regulatory Effect on Ferroptosis in NSCLC

#### 4.3.1. Natural and Extracted Drugs

Currently, bioactive ingredients from Chinese medicine are gradually gaining attention as promising drug candidates in various diseases [83,84,85]. Chen et al. proved the availability of artemisinin in regulating cellular iron homeostasis in human colorectal, breast, and lung cancer cells early. As the cellular iron homeostasis dysfunction related to an occurrence of ferroptosis, it implied an effect of artemisinin towards ferroptosis [86]. Consistent with this, a recent study discovered that artemisinin derivatives ART and DHA worked synergistically to induce ROS-dependent ferroptosis in A549 cell through the downregulation of *xCT* and upregulation of transferrin receptor (*TFRC*), whereas N-Acetyl-L-cysteine (NAC), together with ferrostatin-1, was able to reverse it to some extent [48]. Tian et al. also elucidated the extract from Huaier aqueous could promote ferroptosis by an upregulation of ROS expression level through a series of experiments [49]. Furthermore, high doses of Zinc exhibited an anti-cancer effect in A549 cells, lowering GSH and boosting *GSSG* levels, as previously described. It revealed the fact that Zinc poisoning could cause ferroptosis, and a high-throughput multi-omics analysis, as well as further tests, were used to identify the cause [50].

As curcumin can induce autophagy in NSCLC cells [87,88,89], Tang et al. found it accounted for ferroptosis in A549 and H1299 cells. This ferroptosis was inhibited not just by ferroptosis inhibitor ferrostatin-1 or knocking down the iron-responsive element-binding protein 2 (*IREB2*), but also by autophagy inhibitor chloroquine or Beclin 1 silence. In vitro, the researchers used the cell counting kit-8 (CCK-8) test, *Ki67* immunofluorescence, and transmission electron microscope assays, which they combined with in vivo H&E staining and immunohistochemistry (IHC) studies. These findings verified an existence of autophagy. Then, a corresponding assay kit was also used to detect the intracellular ROS, MDA, superoxide dismutase (SOD), GSH, and iron contents, demonstrating the occurrence of ferroptosis. Moreover, proteins relevant with these two phenotypes were detected through Western blot assay. Finally, the ferroptosis turned out to be triggered by autophagy when NSCLC cells were treated with curcumin [19].

Similarly, Ginkgetin was capable to induce autophagy further triggering ferroptosis. Lou et al. performed cytotoxicity and Western blot assays proving the therapeutic effect of Ginkgetin against NSCLC cells. Then, the lipid peroxidation and labile iron pool were detected. In addition, Western blot assays were also employed, implying the occurrence of ferroptosis in NSCLC cells that was treated with cisplatin. Later, researchers blockaded ferroptosis and used the luciferase assay, immunostaining, chromatin immunoprecipitation, and Annexin V staining to illustrate the specific mechanism of ferroptosis. Finally, in vivo and in vitro tests revealed that when NSCLC cells were treated with Ginkgetin, the ferroptosis-related biomarkers SLC7A11 and CPX4 levels dramatically reduced, while the labile iron pool and lipid peroxidation were enhanced. At the same time, Ginkgetin inactivated the NRF2/HO-1 axis and increased the ROS formation due to its disruption on redox hemostasis. Otherwise, restricting ferroptosis resulted in reversion, suggesting that Ginkgetin was essential in initiating ferroptosis, which then strengthened the anticancer effect of cisplatin-treated NSCLC [51].

#### 4.3.2. Chemical and Synthesized Drugs

Despite the availability of natural cures, a great number of chemical and synthetic medications with anti-cancer efficacy in NSCLC and a connection to ferroptosis were discovered. 

Recently, a protein–protein interaction analysis revealed that the dietary additive Ammonium Ferric Citrate (AFC) induces ferroptosis in NSCLC by regulating *GPX4* and promoting oxidative stress damage. Then, it affected the downstream GSS/GSR complex and the GGT family proteins. Simultaneously, using qPCR assay, a reduction in the autophagy regulators’ expression level was also observed. These results documented that AFC displayed a regulatory effect on autophagy and ferroptosis, namely, restraining autophagy and promoting ferroptosis. Thus, it is notable that there is a possible link between the initiation of ferroptosis and the emergence of autophagy [52]. As erastin was first found to be a ferroptosis-inducer [56], it was recently used together with celastrol, a triterpene extracted from the *Tripterygium wilfordii*, inducing ferroptosis. Thereby, it helped ease the side effects of celastrol in NSCLC. At the same time, erastin raised NSCLC cell sensitivity to celastrol through modulating ROS production and mitochondrial function [90]. Otherwise, through the clinical data analysis of Xuanwei area, researchers focused on two differentially expressed proteins, thioredoxin 2 (TXN2) and haptoglobin (HP), which played a vital role in affecting the effects of erastin and suppressing RSL-induced ferroptosis in vitro. Relatively, it also enhanced the tumor growth in vivo [91].

*P53* is a tumor suppressor gene that regulates various cellular process, and abundant proof has implied its induction on ferroptosis during the tumor progression [92,93]. 

Huang et al. performed semiquantitative Western blot and Annexin VFITC/PI staining, discovering that erastin could upregulate the p53 expression levels. Mechanistically, it was due to the erastin-induced ROS, then leading to ferroptosis in NSCLC. To further demonstrate this hypothesis, Western blot and ROS detection assays were used. It was found that when erastin was co-incubated with ROS-cleaner NAC and ATM-kinase-inhibitor KU55933, p53 manifested a high expression level. Most excitingly, the ROS-activated p53 resulted in a ROS accumulation in A549 cell reversely with the downregulation of SLC7A11, a protein that related with ferroptosis. Therefore, the results indicated the existence of a feedback loop among the erastin-induced ROS, p53, and the erastin-induced ferroptosis. To put it another way, the occurrence of ferroptosis was linked to the ROS/p53/SLC7A11 axis in a significant way. Furthermore, the erastin-induced p53 was proven to trigger both ferroptosis and apoptosis by using the caspase inhibitor Z-VAD-FMK [57]. In terms of protein SLC7A11, Yu et al. applied a bioinformatics analysis method to deal with the RNA sequencing data and discovered that when H1299 cells were treated for 12 h with XAV939, a Wnt/-catenin pathway inhibitor, *SLC7A11* expression was also downregulated. Then, it came along with ferroptosis, simultaneously inhibiting the NSCLC progression. Namely, some inner associations between the XAV939 and p53 were supposed to exist, requiring further studies to unravel. In addition, lncRNA MIR503-host-gene (MIR503HG) was observed to be related to *SLC7A11* and could be downregulated by XAV939 [60]. 

In addition, when treated with levobupivacaine, high levels of ROS, iron, and Fe^2+^ were observed in NSCLC cells, implying an existence of ferroptosis. Likewise, p53 was also overexpressed, resulting in the induction of ferroptosis. These findings revealed the underlying mechanism of levobupivacaine-induced ferroptosis to some degree. In addition, a tumorigenicity analysis verified the inhibitory effect of levobupivacaine on the tumor growth in vivo at the same time [61]. More interestingly, auranofin (AF) exerted a significant inhibitory effect on GPX4 in H1299 cells. However, in comparison to other types of H1299 cells, AF could only have an influence on intracellular levels of lipid peroxidation, which finally led to ferroptosis in mutant p53 R273H accumulating isogenic H1299 cells, indicating a different therapeutic effect. [62].

## 5. Ferroptosis Works against Therapy Resistance of NSCLC

As cisplatin is acknowledged as a first-line chemotherapeutic agent [94], early research also demonstrated its potential to induce ferroptosis and apoptosis in A549 and H1299 cells according to the results from methylthiazoltetrazlium dye uptake and RNAi experiments. Otherwise, the findings of the related assay kit revealed that cisplatin reduced GSH levels and inactivated glutathione peroxidase, resulting in ferroptosis. Additionally, a synergetic anti-cancer effect was also found when cisplatin cotreated with erastin [53]. 

In keeping with this discovery, there was abundant evidence indicating the therapeutic effect of erastin against diverse therapy-resistance in NSCLC. 

On the one hand, erastin could attenuate the chemotherapy resistance. Li et al. discovered that both erastin and sorafenib could induce ferroptosis in NSCLC cells with cisplatin resistance by accumulating ROS and inhibiting the NRF2/xCT pathway, indicating that erastin had a therapeutic impact against chemotherapy resistance. Consistent with the facts in vitro, erastin and sorafenib could also suppress the tumor growth in vivo [54]. Furthermore, Liang et al. newly found an erastin analog, PRLX93936, that could cooperate with cisplatin upregulating the ROS, lipid peroxidation, and Fe2+ levels. At the same time, it also downregulated the *GPX4* and *NRF2* expression, synergistically inducing ferroptosis in NSCLC cells. Further results pointed out that NRF2/Keap1 pathway participated in this process, and it was crucial to combat the cisplatin-resistance. Suppressing *K**EAP1* was capable to rescue the influence of *NRF2* when it was silenced in NSCLC cells [58].

On the other hand, erastin showed a therapeutic effect on the radiotherapy resistance by suppressing *GPX4* in the NSCLC cells with radio-resistance. Pan et al. employed the colony formation and Western blot assay, finding that the *GPX4* expression level was reduced in the radioresistant NSCLC cells when it was treated with erastin. Further, the downregulation of GPX4 would next induce ferroptosis, consistent with the results that the observed cell death could only partially be rescued by deferoxamine instead of Z-VAD-FMK or Olaparib [55].

Moreover, as NSCLC gradually came into drug resistance with erastin, Gai et al. found that when Acetaminophen was cotreated with erastin, it elevated the levels of lipid peroxides and attenuated glutathione. Finally, acetaminophen sensitized NSCLC cells to ferroptosis, and this synergistic effect was also confirmed in vivo. In addition, the sharp decline of NRF2 and heme oxygenase-1 expression was observed, which might be a key to explain the specific mechanism of acetaminophen-induced ferroptosis. However, this change could be reversed by bardoxolone methyl [59].

Otherwise, Deng et al. showed that miR-324-3p could directly target *GPX4* and had a similar effect on RSL3, resulting in ferroptosis and cisplatin resistance reversion in A549 cells. The expression level of miR-324-3p was measured using a qRT-PCR assay, and it was discovered to be downregulated. Thus, researchers overexpressed it later in the cisplatin-resistant A549 cells, successfully reversing the chemotherapy-resistance. Additionally, Annexin V/PI staining was also used to detect the death rate of cell. Then, the specific downstream genes were confirmed by the luciferase report and Western blot assay, unravelling a promotion effect of miR-324-3p to cisplatin-induced ferroptosis [44].

Similar to the chemotherapy-resistance, radiotherapy-resistance, and drug-resistance in NSCLC that could be overcome by the induction of ferroptosis, it also offered a strategy in the targeted therapy-resistance. As the upregulation of GPX4 and mTORC1 were observed in Lapatinib resistant NSCLC cells, researchers used commercial kits and rescue experiments elucidating whether inhibiting mTORC1 would be effective to combat it. Later, the results showed a decreased expression level of GPX4 along with other ferroptosis-related genes variation, indicating an induction on it. As a result, silencing mTORC1 accelerated Lapatinib-induced ferroptosis in NSCLC cells and, in the end, reversed Lapatinib resistance in NSCLC. A Xenograft mouse model was also developed, which confirmed that down-regulation of GPX4 promoted the therapeutic impact of Lapatinib, which was consistent with the in vitro results [95]. 

## 6. Conclusions and Discussions

In conclusion, this review mainly focuses on the ferroptosis in NSCLC and summarizes the study progression in recent years. This review not only mentions the possibility of the FOCAD-FAK signal working synergistically with the GSH-independent pathway as a unique mechanism in the progression of ferroptosis in NSCLC, but also explains the association between ferroptosis and NSCLC from different cascades such as bioinformatics, non-coding RNAs, various drugs, and therapy resistance. All the current research shows that ferroptosis can be utilized as a viable treatment option for patients with NSCLC.

Additionally, it is worthwhile to explore the function of ferroptosis on NSCLC immunotherapy. When CD8+ T cells were stimulated by immunotherapy, data showed that they promoted ferroptosis and, in turn, improved the efficacy of immunotherapy [96]. To be clear, there is currently a significant dearth of comparative experimental investigations on whether ferroptosis can play an important role in improving NSCLC immunotherapy resistance. In the recent time, as the nanoparticles gradually applied into cancer treatment and showed a massive therapeutic potential, it could affect the development of tumor through ferroptosis. Meanwhile, the effect of iron-based nanoparticles on ferroptosis could be achieved by catalyzing the Fenton reaction, which further showed a value to immunotherapy [97,98,99]. These findings imply a potential for nanoparticles in ferroptosis-based immunotherapy, and offer a novel strategy that may be suitable for the clinical NSCLC treatment.

## Figures and Tables

**Figure 1 ijms-22-13335-f001:**
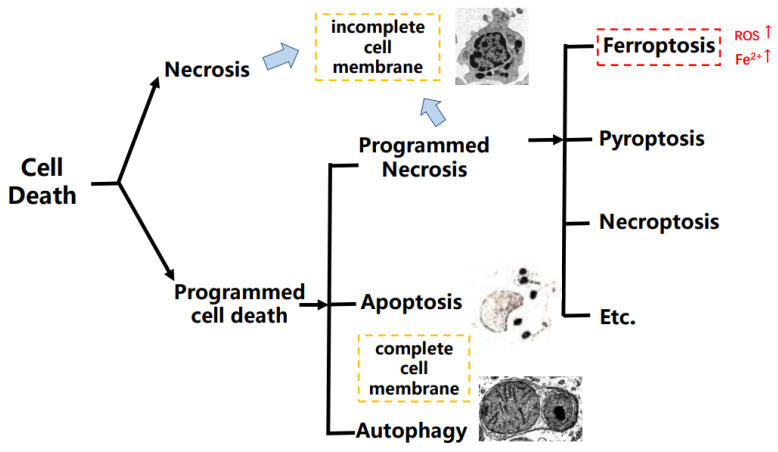
The brief classification on cell death.

**Figure 2 ijms-22-13335-f002:**
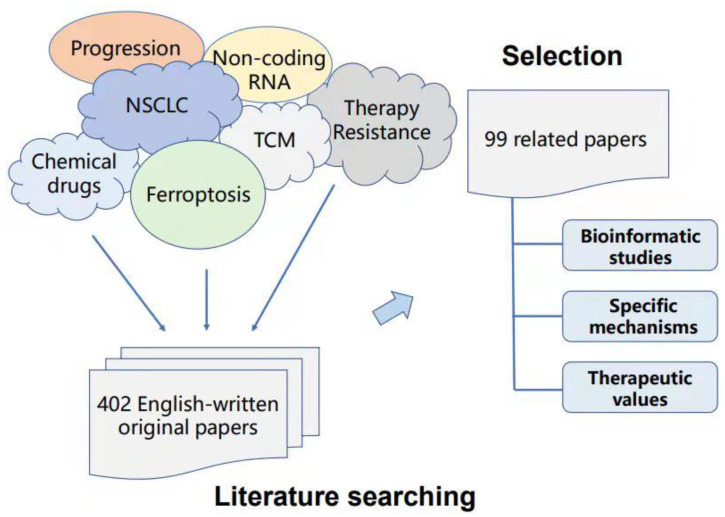
Criteria in this review for article search and selection.

**Figure 3 ijms-22-13335-f003:**
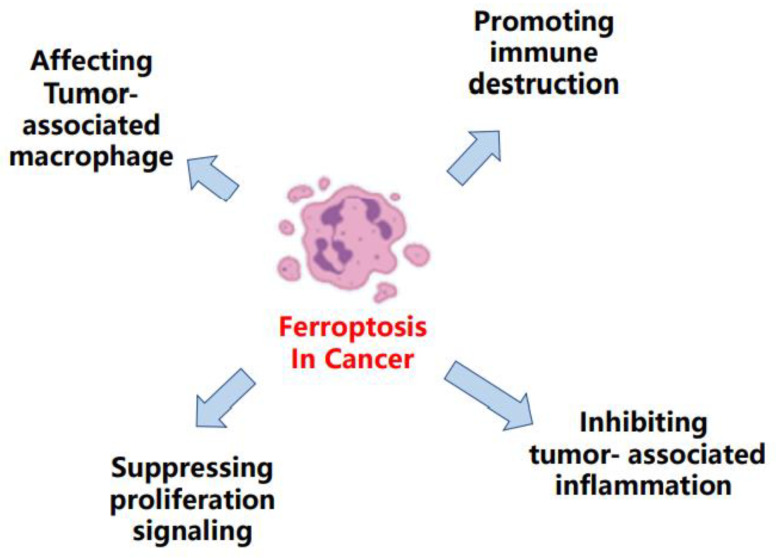
The main molecules regulated by ferroptosis in cancers.

**Figure 4 ijms-22-13335-f004:**
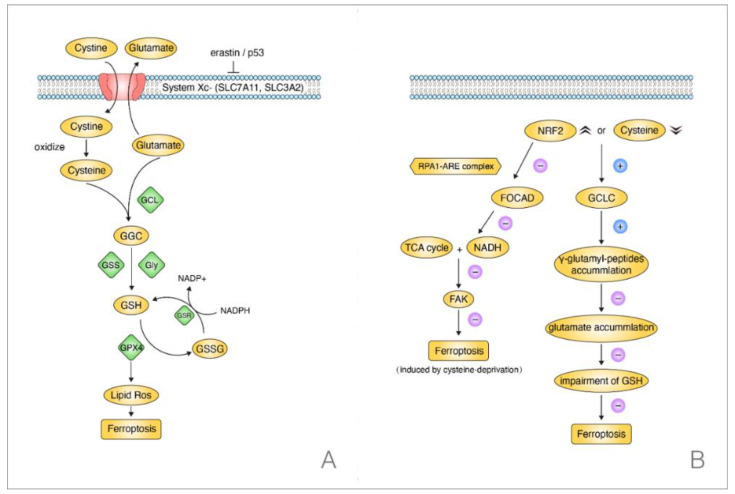
Two regulation pathways concerning ferroptosis in NSCLC when cysteine was silenced. (**A**) GSH-dependent GPX4 signaling, the depleted cysteine induced ferroptosis. (**B**) GSH-independent pathway, the decreased cysteine suppressed ferroptosis without affecting GSH.

**Table 1 ijms-22-13335-t001:** Regulators of ferroptosis progression and mechanism in NSCLC.

Category	Molecular/Drug	Regulation Effect on Ferroptosis in NSCLC	Brief Mechanism	Techniques Used for Detection of Ferroptosis	Function Study	Reference
Micro-RNA	miR-302a-3p	↑	Directly bind ferroportin; Increase the lipid peroxidation and cellular iron level	**_**	In vitro	[42]
	miR-4443	↓	Directly bind METTL3 + Regulate FSP1 in a m6A manner	RT-qPCR; Iron assay kit; Fluorescence staining; Xenograft model; H&E staining; Immunohistochemistry (IHC)	In vitro and In vivo	[43]
	miR-324-3p	↑	Directly bind GPX4, working as a GPX4 inhibitor	Annexin V/PI staining; Western blot; Luciferase report	In vitro	[44]
LncRNA	NEAT1	↑	Bind ACSL4 to Regulate SCL7A11 and GPX4	RT-PCR; Western blot; Dual-luciferase reporter gene assay; Lipid Peroxidation MDA Assay Kit (Beyotime Biotechnology); Iron assay kit	In vitro	[45]
	MT1DP	↑	Negatively regulate NFR2; Increase the Fe2+, ROS, and MDA level; Decrease the GSH level	Dichlorofluorescein diacetate fluorescent probe detection kit; Iron colorimetric assay kit; qRT-PCR; Western blot; xenografts; IHC	In vitro and In vivo	[46]
	LINC00336	↓	Bind ELAVL1; Compete MIR6852 to regulate CBS level	Lipid ROS assays; Iron Assay Kit; RT-qPCR; Immunoblotting;	In vitro	[47]
natural & extracted drugs	Artemisinin derivatives ART and DHA	↑	Downregulate xCT; Upregulate TFRC;	Western blot; RT-qPCR; Reactive oxygen species analysed by flow cytometry	In vitro	[48]
	Extract from Huaier aqueous	↑	Upregulate cellular ROS level	Flow cytometry was used to examine the ROS level; Western blot	In vitro	[49]
	Zinc (high concentration)	↑	Decrease GSH; Increase GSSG	Fluorescence staining	In vitro	[50]
	Curcumin	↑	Ferroptosis is triggered by autophagy; Ferrostatin-1, chloroquine, and silencing IREB2 or Beclin1 could all reverse it.	Intracellular ROS, GSH, and iron contents were determined by corresponding assay kit	In vitro and In vivo	[19]
	Ginkgetin	↑	Ferroptosis is triggered by autophagy; Decrease SLC7A11 and GPX4 levels; Inactivate the NRF2/HO-1 axis + Upregulate the ROS formation.	lipid peroxidation assay; Labile iron pool assay; Western blot; qPCR	In vitro and In vivo	[51]
chemical & synthesized drugs	Ammonium Ferric Citrate	↑	Decrease GPX4; Promote the oxidative stress injury; Ferroptosis is triggered by autophagy	qPCR; ROS detection kit	In vitro	[52]
	Cisplatin	↑	Decrease the GSH level + Inactivate the glutathione peroxidase in A549 and H1299 cells	Intracellular ROS, GSH, and iron contents were determined by the related assay kit	In vitro	[53]
	Erastin	↑	Suppress the NRF2/xCT pathway in the NSCLC cells possessing cisplatin-resistance	RT-qPCR; Western blot; ROS detected by a FACSCalibur Flow Cytometer; Xenograft assay	In vitro and In vivo	[54]
		↑	Inhibit GPX4 in the NSCLC cells owning radio-resistance	Western blot	In vitro	[55]
		↑	Affect the ROS generation and mitochondria when Cotreated with celastrol.	ROS, iron were detected by a FACSCalibur Flow Cytometer; RT-PCR; Western blot; Lipid peroxidation assay kit; Commercial GSH quantification kit	In vitro	[56]
		↑	A feedback loop among the erastin-induced ROS, p53, and the erastin-induced ferroptosis in A549 cell; Increase ROS + Downregulate SLC7A11	semiquantitative Western blot; ROS detection kit	In vitro	[57]
	3-(2-ethoxyphenyl)-2-(piperazin-1-ylmethyl)quinazolin-4(3H)-one	↑	Upregulate the ROS, lipid peroxidation, and Fe2+ levels when Cotreated with cisplatin; Downregulate the GPX4 and NRF2 expression; Regulate the NRF2/Keap1 pathway to avoid the cisplatin-resistance.	**_**	In vitro	[58]
	Acetaminophen	↑	Decrease the NRF2 and heme oxygenase-1 expression	**_**	In vitro and In vivo	[59]
	2-(4-(trifluoromethyl)phenyl)-7,8-dihydro-5H-thiopyrano[4,3-d]pyrimidin-4-ol	↑	Decrease the SLC7A11 level in H1299 cell	RNA sequencing; Gene enrichment analysis	In vitro	[60]
	levobupivacaine	↑	Induce high levels of ROS, iron, and Fe2+; Upregulate p53 to trigger ferroptosis	The reactive oxygen species levels were detected using flow cytometry analysis; Iron Assay Kit; Xenograft model	In vitro and In vivo	[61]
	Auranofin	↑	Inhibit GPX4 in the mutant p53 R273H accumulating isogenic H1299 cell	GSH/GSSG-Glo™ Assay kit; Western blot; Fluorescence staining	In vitro	[62]

## Data Availability

Not applicable.

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
