# Peer review of "Ferroptosis in Non-Small Cell Lung Cancer: Progression and Therapeutic Potential on It"

_ijms, 2021, doi:10.3390/ijms222413335_

Round 1

Reviewer 1 Report

The paper entitled: “Ferroptosis in Non-small Cell Lung Cancer:progression and  therapeutic potential on it” by Jiayu Zou, resulted newly, well written and clear.

Some minor points could improve the paper:

1 add a figure that summarized the main role of ferroptosis and the pathogenetic process in cancer development and progression.

2 I also suggest to the author to consider in this review to add the method of selection of the relevant paper used for the review

3 what are the main methods used to analysed ferroptosis? I also suggest to the author to added the main findings and what are the techniques used for detection of ferroptosis (microscopy, flow cytometry…)

4 I also suggest to the authors to add a paragraph that explain the relevance of ferroptosis an NSCLC than other type of lung cancer or lung diseases.

5 add a table that comprise the list of reference evaluated for the study.

Author Response

The paper entitled: “Ferroptosis in Non-small Cell Lung Cancer:progression and  therapeutic potential on it” by Jiayu Zou, resulted newly, well written and clear.

A:Thanks ever so much for the reviewer. These comments and suggestions are so important for us to improve our work.

Some minor points could improve the paper:

1 add a figure that summarized the main role of ferroptosis and the pathogenetic process in cancer development and progression.

A: Great thanks to the reviewer. We have already added the corresponding figure as Figure 3 in the manuscript according to the vital comment. (line 74-76)

2 I also suggest to the author to consider in this review to add the method of selection of the relevant paper used for the review (figure)

A: Sincere thanks to the reviewer. We have already added the corresponding figure as Figure 2 in the manuscript according to the essential suggestion, and the corresponding content in line 54-63.

3 what are the main methods used to analysed ferroptosis? I also suggest to the author to added the main findings and what are the techniques used for detection of ferroptosis (microscopy, flow cytometry…)

A: We are very grateful for the helpful advice from the reviewer. The corresponding content about techniques used for detection of ferroptosis has been added in Table 1.

4 I also suggest to the authors to add a paragraph that explain the relevance of ferroptosis an NSCLC than other type of lung cancer or lung diseases.

A:Thanks for the essential comments from the reviewer. We have already added the corresponding content in line 123-128 to illustrate the importance and value for summarizing and exploring the role of ferroptosis in NSCLC.

5 add a table that comprise the list of reference evaluated for the study.

A: We really appreciate the critical comments from the reviewer. Anyway, since the clinical studies about targeting ferroptosis in NSCLC are still lacking, the whole articles collected are preclinical studies. In this situation, we are extremely sorry that it is hard for us to list inclusion and exclusion characteristics as the list of reference evaluated for the study. We have collected articles about regulators of ferroptosis in NSCLC with exact function studies. The corresponding content about function study has been added in Table 1. 

Reviewer 2 Report

A review of the importance of ferroptosis in non-small lung cancer is well timed, however, the authors fail to have a conducting logic thought the manuscript, which makes it har d to read and follow.

Specific comments:

  • In several places the authors describe an experiment or some data without providing hard numbers when they are available. For example in line 21 instead of “bad prognosis” the authors should say that lung cancer as a 19% survival rate at 5 years.
  • The introduction is too vague in the description of the programed necrosis, including ferroptosis. This could be addressed possible with a figure, or simple descriptions of each of the programed necrosis pathways.
  • The text has a lot of grammatical inconstancies, and terms that are confusing (such as deprivation, prohibited, depressed).
  • The use of commercial drug names should be avoided.
  • Section 3.2 is very confusing, and the authors fail to convey the main results from each study referred.
  • Section 4.1 the authors should relay the conclusions and/or specific results if necessary for the conducting message of the manuscript, for each bioinformatic study.
  • Table is never referred in the text.

Author Response

A review of the importance of ferroptosis in non-small lung cancer is well timed, however, the authors fail to have a conducting logic thought the manuscript, which makes it hard to read and follow.

A: We really appreciate the critical comments and valuable suggestions from the reviewer. They are so important for us to improve our work. We have added the exact ratio of lung cancer and NSCLC, the description of PCD, the selection of the relevant paper, the techniques used for detection of ferroptosis, more specific relevance of ferroptosis in NSCLC, a figure that summarized the main role of ferroptosis and reorganized the contract to make the manuscript more logic. 

Specific comments:

1.In several places the authors describe an experiment or some data without providing hard numbers when they are available. For example in line 21 instead of “bad prognosis” the authors should say that lung cancer as a 19% survival rate at 5 years.

A: Great thanks to the reviewer. The comment is so essential for the work. We are very sorry about our mistakes without providing the exact number for incidence, mortality and prognosis. We have added the corresponding content in abstract and line 98-103.  

2.The introduction is too vague in the description of the programed necrosis, including ferroptosis. This could be addressed possible with a figure, or simple descriptions of each of the programed necrosis pathways.

A: Sincere thanks to the reviewer. The comment is really helpful for the improvement of the work. The corresponding content has been added in line 49-52 Besides, we also added a figure as Figure 1 to illustrate the corresponding content.

3.The text has a lot of grammatical inconstancies, and terms that are confusing (such as deprivation, prohibited, depressed).

A: Thank the reviewer ever so much. We are so sorry for these mistakes. The whole content has been double-checked, and the corresponding terms have been replaced as silence, inhibited, repressed, etc.

4.The use of commercial drug names should be avoided.

A: We are very grateful for the comments. We are extremely sorry for these mistakes. The corresponding content has been revised in the manuscript.

5.Section 3.2 is very confusing, and the authors fail to convey the main results from each study referred.

A: Thanks for the vital comment. We are so sorry for that. We have added more content to illustrate it and reorganize the section 3.2.

6.Section 4.1 the authors should relay the conclusions and/or specific results if necessary for the conducting message of the manuscript, for each bioinformatic study.

A: Thank the reviewer very much. We have added the specific results on the relative genes for each bioinformatic study in line 204-205 and 219-220.